# Proteomic analysis identifies novel binding partners of BAP1

**Roy Baas[1], Fenna J. van der Wal[2], Onno B. Bleijerveld[3], Haico van Attikum[2], Titia K. Sixma[1]***

**1** Division of Biochemistry and Oncode Institute, Netherlands Cancer Institute, Amsterdam, The Netherlands, **2** Department of Human Genetics, Leiden University Medical Center, Leiden, The Netherlands, **3** Proteomics Facility, Netherlands Cancer Institute, Amsterdam, The Netherlands

* t.sixma@nki.nl

## Abstract

BRCA1-associated protein 1 (BAP1) is a tumor suppressor and its loss can result in mesothelioma, uveal and cutaneous melanoma, clear cell renal cell carcinoma and bladder cancer. BAP1 is a deubiquitinating enzyme of the UCH class that has been implicated in various cellular processes like cell growth, cell cycle progression, ferroptosis, DNA damage response and ER metabolic stress response. ASXL proteins activate BAP1 by forming the polycomb repressive deubiquitinase (PR-DUB) complex which acts on H2AK119ub1. Besides the ASXL proteins, BAP1 is known to interact with an established set of additional proteins. Here, we identify novel BAP1 interacting proteins in the cytoplasm by expressing GFP-tagged BAP1 in an endogenous BAP1 deficient cell line using affinity purification followed by mass spectrometry (AP-MS) analysis. Among these novel interacting proteins are Histone acetyltransferase 1 (HAT1) and all subunits of the heptameric coat protein complex I (COPI) that is involved in vesicle formation and protein cargo binding and sorting. We validate that the HAT1 and COPI interactions occur at endogenous levels but find that this interaction with COPI is not mediated through the C-terminal KxKxx cargo sorting signals of the COPI complex.

## Introduction

Ubiquitination of proteins can have a big impact on protein behavior and lifetime by affecting protein degradation, translocation and conformational change resulting in altered cellular homeostasis. Because of this major impact, protein ubiquitination and deubiquitination needs to be carefully controlled by the cell. Removal of ubiquitin is facilitated by deubiquitinating enzymes (DUBs) that are frequently found in multi-protein complexes with various compositions that influence the DUB activity by activating or repressing the catalytic subunit or recruit the complex to or from sites where DUB activity is required.

BRCA1-associated protein 1 (BAP1) is a polycomb repressive protein that belongs to the UCH class of DUBs. Upon recruitment to polycomb repressive elements on the genome it can deubiquitinate H2AK119ub1 via its catalytic cysteine C91, thereby affecting gene expression [1]. BAP1 in itself has little catalytic activity but becomes activated by ASXL1 or its paralogs ASXL2 or ASXL3 (here further referred to as ASXL), forming a polycomb repressive

**Data Availability Statement:** The mass spectrometry proteomics data have been deposited to the ProteomeXchange Consortium via the PRIDE partner repository with the dataset identifier PXD023676.

**Funding:** KWF Kankerbestrijding (DCS):Titia K Sixma 2015-8082; TKS & HvA; www.kwf.nl Oncode Institute; TKS; www.oncode.nl NWO | Aard- en Levenswetenschappen, Nederlandse Organisatie voor Wetenschappelijk Onderzoek (NWO-ALW OPEN): 2015.091; TKS; www.nwo.nl Nederlandse Organisatie voor Wetenschappelijk Onderzoek (NWO): X-omics Initiative; NKI; www.nwo.nl The funders had no role in study design, data collection and analysis, decision to publish, or preparation of the manuscript.

**Competing interests:** The authors have declared that no competing interests exist.

deubiquitinase (PR-DUB) complex [1] and increasing its $K_M$ [2]. Interaction of ASXL with BAP1 is mediated through the C-terminal ULD domain on BAP1 and the C-terminal DEU-BAD domain on ASXL [2, 3]. The C-terminus of BAP1 is also required for its nuclear localization via the nuclear localization domain and recruitment to nucleosomes.

BAP1 is a tumor suppressor protein [4] that is encoded on chromosome 3p21. Loss of BAP1 is implicated in a distinct subset of cancers like development of mesothelioma [5, 6], uveal and cutaneous melanoma [7, 8], clear cell renal cell carcinoma [9, 10] and bladder cancer [11]. Inactivation of BAP1 causes apoptosis in mouse ES cells, fibroblasts, liver and pancreatic tissue but not melanocytes and mesothelial cells [12]. Its cell type and hence tumor specificity could be partially due to its transcriptional repressive function. The exact mechanism why different cell types behave differently in absence of BAP1 is mostly unknown, but at least in part it involves nucleosome ubiquitination by polycomb complex 1. In the absence of BAP1 in certain cell types, the ubiquitin ligase RING1B (RNF2)in polycomb complex 1 subtypes promotes apoptosis by ubiquitination of H2AK119 on *Bcl2* and *Mcl1* prosurvival genes, resulting in repression of these survival factors [12]. In the non-apoptotic melanocytes (where BAP1 is often found mutated but not the actual driver) RNF2 does not regulate the *Bcl2* and *Mcl1* genes but instead the prosurvival gene *Mitf* becomes expressed upon loss of BAP1 resulting in cell survival [12]. Overall, many questions remain on the mechanisms by which BAP1 affects cell homeostasis, cell fate, proliferation and survival. Some of these functions may be related to the proteins that BAP1 interacts with.

A common approach to study protein interaction partners is by affinity purification of tagged proteins, followed by mass spectrometry (AP-MS) analysis. Most of these studies are done with tagged exogenous proteins alongside its untagged endogenous variant. Using such techniques, previous interaction studies have identified and validated a series of BAP1 interaction partners, including ASXL1/2, FOXK1/2, HCFC1/2, OGT, MBD5/6 and UBE2O [13–19], linking BAP1 to various cellular processes. Additionally other proteins like RBBP7 and HAT1 have been seen interacting with BAP1 but these were never validated [14]. Most processes are regulated via its nuclear fraction, like cell growth [16], cell cycle progression [18], ferroptosis [20, 21], DNA damage response [22, 23] and ER metabolic stress response [24]. However, BAP1 also seems to regulate processes via its cytoplasmic fraction such as promoting apoptosis via modulation of IP3R3 mediated ER $Ca^{2+}$ release [25]. BAP1 itself is regulated by ubiquitination through the E2 ubiquitin conjugating enzyme UBE2O [26]. Ubiquitination of BAP1 by UBE2O results in sequestration of BAP1 in the cytoplasm, rendering it unable to find its nuclear targets. In turn, BAP1 regulates its level of ubiquitination via auto-deubiquitination [26]. Additional regulation of BAP1 is mediated through the monoubiquitination of activating ASXL1 or ASXL2 proteins by UBE2E family of proteins, resulting in stabilization of the ASXL protein [27].

The vesicle coat protein complex I (COPI) complex is composed of seven proteins that can be biochemically dissected in a Cage/B-subcomplex (α-COP, β'-COP and ε-COP) and an Adapter/trunk/F-subcomplex (β-COP, δ-COP, γ-COP and ζ-COP) [28]. Together, these seven subunits form the cytoplasmic heptameric coatomer complex. The main described functions of COPI are vesicle formation and cargo sorting and binding. COPI is the central protein complex that facilitates the Golgi to ER transport and intra-Golgi transport, while other membrane associated functions have also been described (for COPI reviews see [28, 29]). Upon vesicle formation, GTP-bound ARF1 inserts a myristoylated N-terminal amphipathic helix into the lipid bilayer to which the COPI heptamer is *en bloc* recruited. Cargo protein binding by COPI is mediated by signals found on the respective cargo proteins. Cargo proteins carrying C-terminal KKxx and KxKxx motifs are bound by the N-terminal WD40-repeats of α-COP and β'-COP respectively [30]. Placement of the lysines at the -3 and either -4 or -5 position from the C-terminus is critical [31].

Here, we identify novel BAP1 binding partners in the cytoplasmic fraction of cells expressing GFP-tagged BAP1 in absence of endogenous BAP1 protein. Removal of endogenous BAP1 enhances the ability to identify binding partners, as all BAP1 is tagged and therefore can be retrieved with interacting proteins. AP-MS analysis on these cells identify the full heptameric COPI complex along other proteins as novel binding partners. These interactions are validated on an endogenous level.

## Materials and methods

### Plasmids and cloning

The vectors pcDNA5.1-FRT/TO-puro-(N)GFP-TEV-FLAG-3C-LIC, pX330, pBABE-puro and pcDNA5-FRT/TO-puro-eGFP-NLS were available in-house, BAP1 cDNA [2] was kindly provided by dr. Jürg Müller (for generation of pcDNA5.1-FRT/TO-puro-(N)GFP-TEV-FLAG-3C-BAP1) or originated from Wade Harper's laboratory obtained from Addgene (for generation of pcDNA5-FRT/TO-puro-eGFP-BAP1). Primers and gBlock Gene Fragments were purchased at Integrated DNA Technologies. All obtained and cloned DNA constructs were sequence verified using Sanger sequencing.

### Primers and gBlock Gene Fragments

| LIC cloning Primers | |
| --- | --- |
| BAP1 LIC N Fw | CAGGGACCCGGTAATAAGGGCTGGCTGGAGCTG |
| BAP1 LIC N Rv | CGAGGAGAAGCCCGGTCACTGGCGCTTGGCCTTG |
| COPE LIC N Fw | CAGGGACCCGGTGCTCCGCCAGCGCCCGGC |
| COPE LIC N Rv | CGAGGAGAAGCCCGGTCATGCGGAAGGCGCGTATTGCAGGACCAG |
| Mutagenesis primers | |
| N stop Fw | TTGACCGGGCTTCTCCTCGAGTC |
| N stop Rv | CCGGGTCCCTGAAAGAGCAC |
| BAP1 670 Rv | GTTGTGGGTCCTTCTCTGGTC |
| BAP1 stop Fw | TGACCGGGCTTCTCCTCGAG |
| BAP1 711 Fw | AAGCAGCGGAAGCCTGACC |
| BAP1 SASRQ Fw | CGCCAGTGACCGGGCTTCTC |
| BAP1 SASRQ Rv | ACTGGCACTGTAGGGGCGAG |
| BAP1 C91S Fw | TACCCAACTCTTCTGCAACTCATGCC |
| BAP1 C91S Rv | GGCATGAGTTGCAGAAGAGTTGGGTA |
| BAP1 intron 3 Fw | GAGACTGGTGTGGGTGTTCA |
| BAP1 intron 4 Rv | CAGTTCGTTCTGCCAGAGGAT |
| gBlock Gene Fragments | |
| COPE (E.coli codon optimized) | CAGGGACCCGGTGCTCCGCCAGCGCCCGGCCCAGCTAGCGGTGGATCGGGTGAGGTGGACGAATTGTTCGACGTGAAAAATGCATTCTATATCGGAAGTTATCAGCAGTGCATCAACGAGGCGCAACGTGTTAAGTTGTCGAGTCCTGAGCGCGATGTGGAGCGCGATGTATTTCTGTATCGCGCGTATTTGGCACAGCGCAAATTTGGAGTGGTTTTTGGATGAAATTAAGCCTTCCAGTGCGCCGGAACTGCAGGCGGTTCGCATGTTTGCGGACTACCTGGCCCATGAAAGCCGTCGTGATAGCATTGTAGCCGAGCTTGACCGCGAGATGAGTCGCAGTGTGGATGTAACCAATACTACTTTTCTGCTTATGGCCGCATCAATTTATTTACACGACCAGAACCCGGACGCAGCATTACGCGCCCTTCATCAAGGAGATTCGCTTGAATGCACCGCCATGACCGTGCAAATTTTACTTAAATTAGACCGCTTAGACCTTGCCCGCAAAGAGTTAAAGCGCATGCAGGATCTTGATGAGGATGCCACTCTGACACGATTAGCCGGAGGTGGGTTAGCTTGGCCCACTGGAGGGGAGAAACTTCAAGACGCATACTATATCTTTCAAGAGATGGCGGATAAGTGTAGTCCCACATTACTGCTGTTAAACGGTCAGGCGGCCTGTCACATGGCACAGGGGCGTTGGGAAGCAGCTGAAGGGTTGCTGCAAGAGGCGCTTGATAAAGACTCTGGGTACCCAGAGACCTTAGTTAACTTGATTGTATTGTCGCAGCATTTAGGCAAACCCCCTGAGGTGACGAATCGCTATCTTTCACAGCTGAAAGATGCTCATCGTAGTCACCCGTTCATTAAGGAATATCAGGCTAAAGAGAACGACTTCGACCGTCTGGTCCTGCAATACGCGCCTTCCGCATGACCGGGCTTCTCCTCG |
| CRISPR guides | |
| BAP1 exon 4 guide Fw | CACCGCCGGCGAAAGGTCTCTACCT |
| BAP1 exon 4 guide Rv | AAACAGGTAGAGACCTTTCGCCGG |

## BAP1 and COPE ligation independent cloning (LIC)

BAP1 cDNA was amplified with LIC overhang sites using the BAP1 LIC N Fw and BAP1 LIC N Rv primer pairs and the BAP1 cDNA [2] as template. For the PCR the Phusion Flash High-Fidelity PCR Master Mix was used with a final volume of 20 μL. PCR program was composed of an initial melting step of 10 seconds at 98˚C, followed by 35 repetitions of 1 sec melting step at 98˚C, 5 seconds annealing step at 60˚C and 1 minute elongation step at 72˚C. Final steps were an elongation step of 2 minutes at 72˚C and cooling of the sample at 12˚C until further processing.

COPE cDNA was amplified with LIC overhang sites using the COPE LIC N Fw and COPE LIC N Rv primer pairs and the COPE gBlock as template. For the PCR the Phusion Flash High-Fidelity PCR Master Mix was used with a final volume of 50 μL. PCR program was composed of an initial melting step of 3 minutes at 98˚C, followed by 35 repetitions of 20 sec melting step at 98˚C, 10 seconds annealing step at 60–72˚C and 20 second elongation step at 72˚C. Final steps were an elongation step of 1 minute at 72˚C and cooling of the sample at 12˚C until further processing.

LIC cloning was performed as described before [32]. Cloning vector pcDNA5.1-FRT/TO-puro-(N)GFP-TEV-FLAG-3C-LIC was KpnI digested and 400 ng was used with 100 ng BAP1 or COPE PCR insert for T4 DNA polymerase treatment in presence of 1x NEB buffer 2 and 25 mM dTTP or dATP in a total volume of 10 μL or 20 μL for the vector and insert respectively. Samples were incubated for 30 minutes at room temperature to generate LIC single-strand overhang. Enzymes were inactivated for 20 minutes at 75˚C. 2 μL treated insert and 1 μL treated vector were combined and transformed into E.coli strain DH5α. Individual colonies were picked and resulting pcDNA5.1-FRT/TO-puro-(N)GFP-TEV-FLAG-3C-BAP1 and pcDNA5.1-FRT/TO-puro-(N)GFP-TEV-FLAG-3C-COPE were isolated.

## BAP1 truncation and C-terminal tail mutants cloning

The pcDNA5.1-FRT/TO-puro-(N)GFP-TEV-FLAG-3C-BAP1 2–760, BAP1 Δ671–710, N-stop and BAP1 SASRQ constructs were generated using PCR reactions. BAP1 2–670 was generated using BAP1 670 Rv and N stop Fw primers on pcDNA5.1-FRT/TO-puro-(N)GFP-TEV-FLAG-3C-BAP1 template. BAP1 Δ671–710 was generated using BAP1 670 Rv and BAP1 711 Fw primers on pcDNA5.1-FRT/TO-puro-(N)GFP-TEV-FLAG-3C-BAP1 template. N-stop was generated using N stop Fw and N stop Rv primers on pcDNA5.1-FRT/TO-puro-(N)GFP-TEV-FLAG-3C-LIC template. BAP1 SASRQ was generated using BAP1 SASRQ Rv and BAP1 SASRQ Fw primers on pcDNA5.1-FRT/TO-puro-(N)GFP-TEV-FLAG-3C-BAP1 template. PCR reactions for BAP1 2–760, BAP1 Δ671–710 and N-stop used the same program composed of an initial melting step of 3 minutes at 98˚C, followed by 35 repetitions of 20 sec melting step at 98˚C, 10 seconds annealing gradient step at 55–65˚C and 3 minute elongation step at 72˚C. BAP1 SASRQ had a similar program with the exception that a 10 seconds annealing gradient step at 60–72˚C was used. Final steps were an elongation step of 5 minutes at 72˚C and cooling of the sample at 12˚C until further processing. A 10% fraction of the PCR samples was used to analyze on a 0.8% agarose gel and fractions to be further processed were selected. Remaining PCR product was DPNI digested to remove template DNA and samples were run on a 0.8% agarose gel. The bands containing the vectors were excised and gel extracted. Samples were dried using speedvac. DNA was dissolved in 8 μL MQ and supplemented with 1 μL 10x DNA ligase buffer and 1 μL T4 PNK kinase and incubated for 30 minutes at 37˚C for phosphorylation. Samples were ligated by addition of 1μL T4 ligase and 40 minutes incubation at room temperature.

For the immunoprecipitation experiments showing the BAP1-HAT1 interaction, pcDNA5-FRT-TO-puro-eGFP-BAP1 was made with an in-house vector containing pcDNA5-FRT-TO-puro-eGFP and FLAG-tagged cDNA of BAP1, using the restriction sites BsrGI and Acc65I. The pcDNA5-FRT-TO-puro-eGFP-BAP1-C91S vector was made with site-directed mutagenesis, using the BAP1 C91S Fw and BAP1 C91S Rv primers and the QuickChange Lightning Multi Site-Directed Mutagenesis Kit form Agilent. The mixture was incubated with 1 μL DpnI for 1 hour at 37˚C. All constructs were transformed into E.coli strain DH5α. Individual colonies were picked and resulting constructs were isolated.

## Cell culture

HeLa FRT cells were kindly provided by dr. Geert Kops. U2OS cells were kindly provided by dr. Jiri Lukas. HeLa FRT and U2OS cells were grown in Dulbecco's Modified Eagle's Medium (DMEM) containing 4.5 g/L glucose and supplemented with 10% v/v fetal bovine serum, 10 mM L-glutamine (HeLa) or GlutaMAX (U2OS), 100 U/mL penicillin and 100 U/mL streptomycin. HeLa FRT wild-type or ΔBAP1 cells were cultured in medium supplemented with 5 μg/mL blasticidin and 200 μg/mL zeocin. FRT-mediated recombined cells were cultured in medium supplemented with 5 μg/mL blasticidin and 1 μg/mL puromycin.

## Polyclonal cell line generation

HeLa FRT wild-type or ΔBAP1 cells were seeded in a 6 wells plate at a cell density of 300 000 cells per well in medium without blasticidin and zeocin 8 - 24h prior to transfection. Cells were co-transfected with 2 μg pcDNA5.1-FRT/TO-(N)GFP-TEV-FLAG-3C expression vector and 0.2 μg pOG44 Flp recombinase expression vector using polyethyleneimine (PEI). Growth medium was replaced 48h after transfection with selection medium and surviving cells were allowed to repopulate the well.

## BAP1 knockout cell line generation using CRISPR

Guides against BAP1 were designed online (crispr.mit.edu) and cloned into the pX330 vector via the BbsI restriction sites. HeLa FRT wild-type cells were seeded in a 6-well plate at a cell density of 300 000 cells per well, 16h prior to transfection. Cells were co-transfected with 3μg pX330-sgBAP1 and 0.5μg pBABE-puro using PEI. Growth medium was replaced 48h after transfection with puromycin selection medium to select for transfected cells. Selection medium was removed and remaining cells were seeded in a 96 well plate at an average cell density of 0.5 cells per well and monoclonal cells were allowed to populate the well. For each clone, genomic DNA was isolated by resuspending cells in SE buffer containing 10 mM Tris-HCl (pH 7.3), 10 mM EDTA and 200 mM NaCl. 10 μl Proteinase K and 25 μl 20% SDS were added and samples were incubated at 55˚C for 3 hours. After addition of 600 μl phenol-chloroform-isoamyl alcohol, vortexing and spinning down at 14000 rpm for 10 minutes, the upper aqueous phase was collected, 1/10 volume 2 M NaAc pH 5.6 and 2 volumes 100% EtOH were added and samples were frozen at -80˚C for 2 hours. After spinning down for 20 minutes at 14000 rpm at 4˚C, pellets were dried at 50˚C and resuspended in TE buffer. PCR was performed on the genomic DNA using Taq PCR Master Mix from Qiagen and BAP1 intron 3 Fw and BAP1 intron 4 Rv. PCR products were sequenced using the BAP1 intron 3 Fw primer. Clone genomic sequences were checked using TIDE [33] and CRISP-ID [34] analysis by comparing them to the wild-type sequence. BAP1 protein levels were checked using immunoblot analysis.

## Confocal microscopy

Cells grown on coverslips were washed with phosphate-buffered saline (PBS) and fixed by incubation with 4% formaldehyde in PBS for 20 minutes at room temperature. Cells were incubated with 2 mg/L DAPI and 1x phalloidin-633 (SC-363796, Santa Cruz Biotechnology) in PBS to stain for nuclei and Tubulin. Coverslips were mounted onto microscopy slides using Immu-Mount (Invitrogen) and left to dry overnight at 4˚C. Images were taken using a Leica TCS SP5 confocal microscope with HCX PLAN Apochromat lambda blue 63.0x1.40 oil UV objective. Samples were scanned using a 405-nm Violet Diode laser for DAPI staining, a 488-nm argon laser for GFP and a 633-nm helium-neon laser for phalloidin. Detector gain (PMT and HyD) and offset (PMT) settings were kept constant within experiments that compared a fusion protein in different conditions.

## Immunoblot analysis

Protein samples were taken up in sample buffer (sample final concentration 40 mM Tris-HCl, pH 6.8, 1% SDS, 5% glycerol, 0.0125% bromophenol blue, 1% β-mercaptoethanol) and incubated for 5 minutes at 95˚C. Proteins were run on a 4–12% SDS-PAGE in MES running buffer for H2A or H2AK119ub analysis or MOPS running buffer for analysis of all other proteins. Samples were transferred to a 0.45 μm nitrocellulose blotting membrane. The membrane was stained using ponceau staining and imaged using the Bio-Rad ChemiDoc XRS+ with Image Lab version 5.1 build 8. Membranes were subsequently developed with the appropriate antibodies and Clarity Western ECL (Bio-Rad) or SuperSignal West Femto Maximum Sensitivity Substrate (Thermo Scientific) and scanned using the Bio-Rad ChemiDoc XRS+ and Image Lab version 5.1 build 8. Immunoblot bands were quantified using Image Lab version 6.0.1 build 34 and visualized using R version 4.0.2 with Rstudio version 1.3.1093. For the immunoprecipitations showing the BAP1-HAT1 interaction, nitrocellulose membranes were developed with appropriate antibodies and scanned on Odyssey Imaging Sytem (LI-COR Biosciences).

## Antibodies

The used antibodies are listed here: BAP1 (C-4, SC-28383, Santa Cruz Biotechnology, all BAP1 immunoblots are stained using this antibody and this antibody is used for the endogenous immunoprecipitation to show the HAT1 interaction), BAP1 (C15200212, Diagenode), BAP1 (A302-243A-T, Bethyl), BAP1 (D1W9B, 13187S, Cell Signaling Technology), Normal mouse IgG (SC-2025, Santa Cruz Biotechnology), Normal rabbit IgG (2729S, Cell Signaling Technology) FLAG (M2, F3165, Sigma-Aldrich), H2A(#07–146, Millipore), H2AK119ub (D27C4, #8240, Cell Signaling Technology), Living Colors (GFP) (JL-8, 632380, Clontech), α-Tubulin (DM1A, CP06, Calbiochem), HAT1 (H-7, SC-390562, Santa Cruz Biotechnology), Goat anti-Rabbit IgG (H+L) Secondary Antibody (1858415, Pierce Biotechnology), Goat anti-Mouse IgG (H+L) Secondary Antibody (1858413, Pierce Biotechnology), CF770 goat anti-mouse IgG (H+L) Highly Cross-Absorbed (#20077, Biotium).

## Crystal violet staining

Cells grown in 6-well plates were washed with PBS and fixed in 4% formaldehyde in PBS for 20 minutes at room temperature. Fixed cells were incubated with crystal violet stain on shaker for 60 minutes at room temperature. Stained cells were washed with demi water until no unbound crystal violet was left. Plates were air-dried and scanned, images were quantified using FIJI.

## Whole cell extract preparation

Confluent cells grown on 15 cm petri dishes were kept on ice and PBS washed twice. Plates were aspirated and 500 μL RIPA buffer (50 mM Tris-HCl pH8.0, 150 mM NaCl, 5mM MgCl2, 10% glycerol and 0.1% IGEPAL CA-630) was added to the plates. Cells were scraped and collected in 1.5 mL Eppendorf tubes. Benzonase was added and incubated on a rotating wheel at 4˚C to remove DNA. Extracts were spun 20 000 g for 15 minutes at 4˚C and soluble fraction was collected as whole cell extract.

## Cytoplasmic and nuclear extract preparation

Roller bottles with a surface area of 2125 cm$^2$ were inoculated with cells from a confluent 15 cm petri dish in a total volume of 160 mL. After 48h, doxycycline was added to the medium to a final concentration of 1μg/mL and left for 24h - 48h to induce protein expression. Roller bottles were washed twice with PBS and cells were collected using trypsin. Cells were collected in 50 mL tubes and washed three times with PBS. Cytoplasmic and nuclear extract preparation and protein concentration determination using Bradford was then performed as described before [35].

## Immunoprecipitation (GFP and endogenous)

Extracts containing GFP-fusion proteins were incubated on GFP-Trap agarose beads (Chromotek) for 20 minutes at 4˚C on a rotating wheel in presence of Ethidium Bromide. After incubation, samples were spun at 1500 g for 2 minutes and supernatant was removed. Samples were washed 3 times using WCE or C300 buffer for WCE or cytoplasmic and nuclear extracts respectively. On the final wash, the final volume was removed using gel loader tips.

For showing the BAP1-HAT1 interaction, the immunoprecipitation was done by collecting the cells overexpressing the GFP-tagged constructs and lysing them in EBC buffer with Benzonase. After 1 hour incubation, samples were spun down 10 min at 14000 rpm and 4˚C and supernatant was added to GFP-Trap agarose beads from Chromotek and incubated for 1.5 hours at 4˚C. After 5 times washing with EBC buffer, beads were diluted in Laemmli buffer (sample final concentration 40 mM Tris-HCl, pH 6.8, 3,35% SDS, 16,5% glycerol, 0.005% bromophenol blue, 0.05 M DTT).

Extracts containing only endogenous proteins were incubated overnight with the preferred antibody at 4˚C on a rotating wheel in presence of Ethidium Bromide. Next day the lysate was transferred to SureBeads Protein G Magnetic beads for 2 hours at 4˚C on a rotating wheel. Samples were washed three times using WCE buffer and a SureBeads Magnetic Rack (Bio-Rad).

For the endogenous BAP1-HAT1 interaction, cells in EBC-MgCl2 buffer with Benzonase for 1 hour at 4˚C. After spinning down 10 min at 14000 rpm and 4˚C, the supernatant was incubated for 3 hours with 2 μg with the preferred antibody, after which Protein G agarose/ Salmon sperm beads (Merck Millipore) were added and incubated for 1 hour. Samples were washed 6 times with EBC buffer before dilution in Laemmli (sample final concentration 40 mM Tris-HCl, pH 6.8, 3,35% SDS, 16,5% glycerol, 0.005% bromophenol blue, 0.05 M DTT).

## Mass spectrometry sample preparation

Mass spectrometry sample preparation was essentially performed as described [35]. Beads containing IP samples were washed three times using PBS to remove residual IGEPAL CA-630 and then completely aspirated using a gel-loader tip. Proteins were alkylated using 2-chloroacetamide (C0267-100G, Sigma-Aldrich) in elution buffer (100 mM Tris-HCl pH 7.5, 2 M urea,

10 mM DTT). Proteins were digested by 2 h Trypsin/Lys-C mix incubation (V5073, Promega) and collected. Beads were eluted again and eluates were combined and Trypsin/Lys-C digested overnight. Tryptic digests were desalted using Stage-Tips [36].

## Mass spectrometry

Tryptic peptides were eluted twice from Stage-Tips using mass spectrometry buffer B (80% acetonitrile, 0.1% formic acid). Peptides were dried and dissolved in 8 μL 10% formic acid of which 3 μL (37.5%) was injected. Tryptic peptides were separated using an EASY-nLC 1000 system (Thermo Scientific) mounted with a C18 analytical column (ReproSil-Pur 120 C18-AQ 2.4 μm (Dr. Maisch GmbH); 75 μm x 500 mm) operating online with an Orbitrap Fusion Tri-brid mass spectrometer (Thermo Scientific). Peptides were eluted from the analytical column at a constant flow of 250 nl/min in a 90-min gradient, containing a 74-min linear increase from 6% to 30% solvent B, followed by a 15-min wash at 100% solvent B. Peptides were fragmented in the ion trap by HCD, the mass spectrometer running in 'top speed' mode with 3 s cycles. The mass spectrometry proteomics data have been deposited to the ProteomeXchange Consortium via the PRIDE partner repository with the dataset identifier PXD023676.

Reviewer account details: username: reviewer_pxd023676@ebi.ac.uk; password: CyuxGkrJ.

## Mass spectrometry data analysis

Proteins were identified from the raw data using MaxQuant [37] version 1.6.0.1 using the Uni-Prot human FASTA database. Carbamidomethylation of cysteines was set as fixed modification, whereas Acetyl (Protein N-term), GlyGly (K) and Oxidation (M) were set as variable modifications with a limit of 5 modifications per peptide. Data were analyzed using Perseus version 1.6.0.7 and visualized using R version 4.0.2 with Rstudio version 1.3.1093.

## Results

### AP-MS analysis of BAP1

BAP1 becomes activated by binding to ASXL proteins to form the PR-DUB complex. Additionally, many different proteins have been identified to bind and interact with BAP1. To study the BAP1 interactome and benchmark these results against existing BAP1 proteome studies [13–15], stable HeLa cell lines that express an inducible (N)GFP-FLAG-BAP1 fusion protein (also referred to as GFP-BAP1) were generated via FRT-mediated recombination, after which its expression and localization were validated (S1 Fig and S1 File).

We generated cytoplasmic and nuclear extracts from a large batch of cells (S2 Fig) and used these to identify interacting proteins by AP-MS (see S3 Fig for quality of reproducibility and cell fractionation). In the cytoplasmic extract GFP-BAP1 pulls down ASXL1/2, FOXK1/2, HCFC1, OGT and UBE2O selectively (Fig 1A). This is a subset of the known BAP1 interacting proteins [13–15, 19]. In the nuclear fraction this set is complemented by KDM1B and MBD6 (Fig 1B). Interestingly, in the volcano plot for the cytoplasmic fraction additional proteins are observed near the edge of the significance curve that are enriched to a lesser degree and with a lower significance score than the previously well-defined interacting proteins. On the right-hand side of this protein cloud are the proteins COPA, COPB2 and ARCN1 that have not been observed in BAP1 AP-MS experiments before. These proteins are of special note because they are enriched 2–4 fold lower than ASXL2 but on the higher end compared to the rest of the interacting proteins and all three are part of the heptameric COPI protein complex. Other proteins in this protein cloud include NPM1, RBBP7 and HAT1, some of which were seen interacting with BAP1 before [14]. To investigate if the proteins in this cloud contain actual BAP1

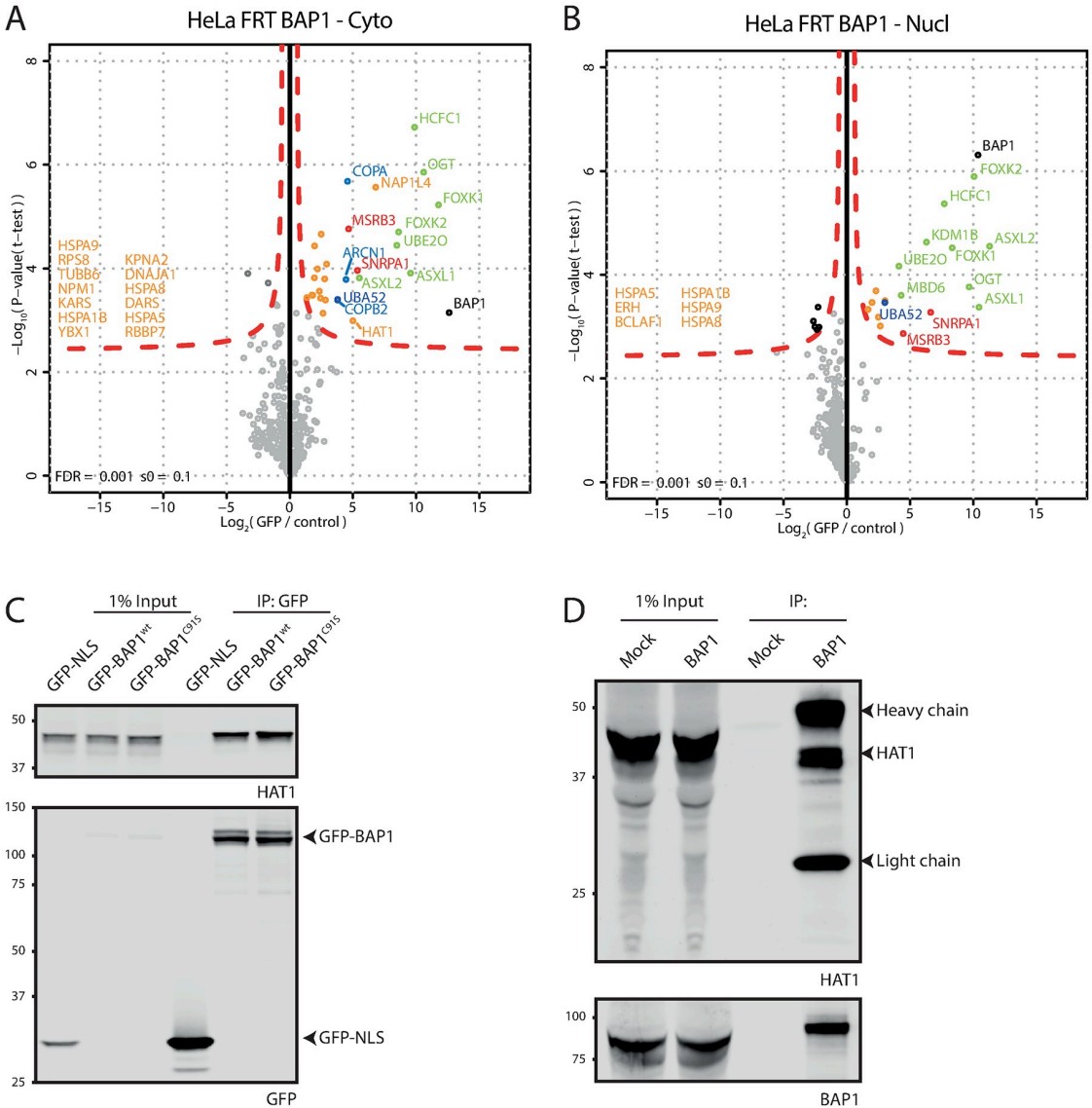

**Fig 1. GFP-BAP1 proteomics show established BAP1 biology and validate HAT1 as a BAP1 interacting protein.** (A and B). Volcanoplots showing GFP-BAP1 interaction partners for cytoplasmic (A) and nuclear fractions (B). (C). GFP-fusion constructs were expressed in HeLa cells and lysates were used in GFP immunoprecipitation. Immunoblot analysis using listed antibodies show HAT1 as a BAP1 interacting protein. (D) U2OS wt cell lysates were used for mock or BAP1 immunoprecipitation. Immunoblot analysis using listed antibodies confirms HAT1 interaction in U2OS cells.

interacting proteins we validated the interaction with the HAT1 protein that is seen near the significance border. A GFP coIP experiment of both GFP-BAP1 and the catalytic-dead BAP1 C91S mutant shows coimmunoprecipitation of HAT1 (Fig 1C). Additionally, an endogenous IP of BAP1 in a different cell line (U2OS) confirmed the HAT1 interaction (Fig 1D) and showed it is not HeLa cell-specific. Overall, these data suggest that HAT1 is a *bona fide* interactor of BAP1 and that the proteins identified near the border of significance in the mass spectrometry experiment are likely real interactors that aren't identified due to random noise in the data that may cause these proteins to become significant.

## Generation of BAP1 mutant cell lines in absence of endogenous BAP1

The BAP1/ASXL PR-DUB complex consists of a BAP1 dimer together with one ASXL molecule [2, 13, 15]. When a tagged BAP1 construct is introduced in HeLa FRT wild-type cells, the tagged construct can dimerize with either another tagged construct or endogenous BAP1. Additionally, endogenous BAP1 dimers will be present that cannot be studied using the AP-MS approach. Importantly, to study the interaction partners of BAP1 truncation variants, dimerization with an endogenous BAP1 wild-type protein must be avoided, because this might yield false positive interactions. To study BAP1 interaction partners of BAP1 truncation constructs, the HeLa FRT ΔBAP1 cell line was generated using CRISPR (Fig 2A and 2B and S4 Fig). Monoclonal HeLa FRT ΔBAP1 clones 4C5 and 4D2 showed loss of BAP1 expression on immunoblot and global H2AK119ub1 levels were elevated 3-fold (Fig 2A and S1 File). Clone 4D2 was selected for further experiments because of the small but out-of-frame deletions in that clone.

BAP1 is activated by binding of ASXL family proteins to its ULD domain [2, 3]. To study BAP1 interactors in the absence of ASXL activation, two different GFP-BAP1 ΔULD constructs were introduced in the 4D2 BAP1 deficient clone (Fig 2C). BAP1 2–670 is a C-terminal truncation construct that has part of the ULD domain deleted and cannot be bound by ASXL [2]. BAP1 Δ671–710 is a ULD domain deletion mutant that cannot bind ASXL but contains the C-terminal extension needed for nuclear localization via the NLS and nucleosome recognition [2]. As controls, 4D2 stable cell lines with BAP1 full-length and the GFP linker followed by a stop codon (N-stop) were also generated. Confocal microscopy analysis shows differential localization for each construct (Fig 3A). BAP1 full-length is mainly nuclear with some cytoplasmic staining and corroborates with earlier data. BAP1 2–670 lacks the nuclear localization signal and localizes mainly to the cytoplasm. Additionally, the intensities of this construct are increased (Fig 3B), this is possibly due to differential regulation of protein stability or turnover compared to the full-length construct. The BAP1 Δ671–710 construct has similar localization as BAP1 full-length due to the presence of the nuclear localization signal in the C-terminal extension, however intensity levels are increased, again suggesting differential regulation compared to full-length. Taken together these data suggest that the linker does not affect protein localization or regulation.

## AP-MS analysis of BAP1 mutation cell lines shows novel interacting proteins

Cytoplasmic and nuclear extracts were generated from the BAP1 full-length, 2–670, Δ671–710 and N-stop cell lines. Immunoblot analysis shows difference in construct size and intensity distribution over both extracts, corroborating with the confocal microscopy data (S4 Fig). AP-MS analysis of BAP1 full-length expressing cell extracts shows identification of ASXL1/2, FOXK1/2, HCFC1/2, OGT and UBE2O, the same set as observed before in cytoplasmic and nuclear extracts for full-length BAP1 (Fig 4 and S5 Fig). An additional cloud of interacting proteins is observed in the cytoplasmic fraction that is markedly absent in the nuclear fraction of the same cell line. In this set of interaction proteins we again found HAT1 and RBBP7. It also included ANKHD1, CBX3, ANKRD17 which have been identified as BAP1 interactors before [14]. Strikingly, all seven subunits (COPA [α-COP], COPB1 [β-COP], COPB2 [β'-COP], COPG1 [γ-COP], ARCN1 [δ-COP], COPE [ε-COP] and COPZ1 [ζ-COP]) that make up the COPI complex are enriched as interacting proteins. Another set of interesting proteins include the NAP1L1, NAP1L4, NAP1L5, C1QBP, MAGED2 and NPM proteins, these proteins are often seen interacting together [38]. Interaction with AHCYL1 is also observed, this may be linked to the IP3R3 regulation by BAP1 because AHCYL1 (which is also known as IRBIT)

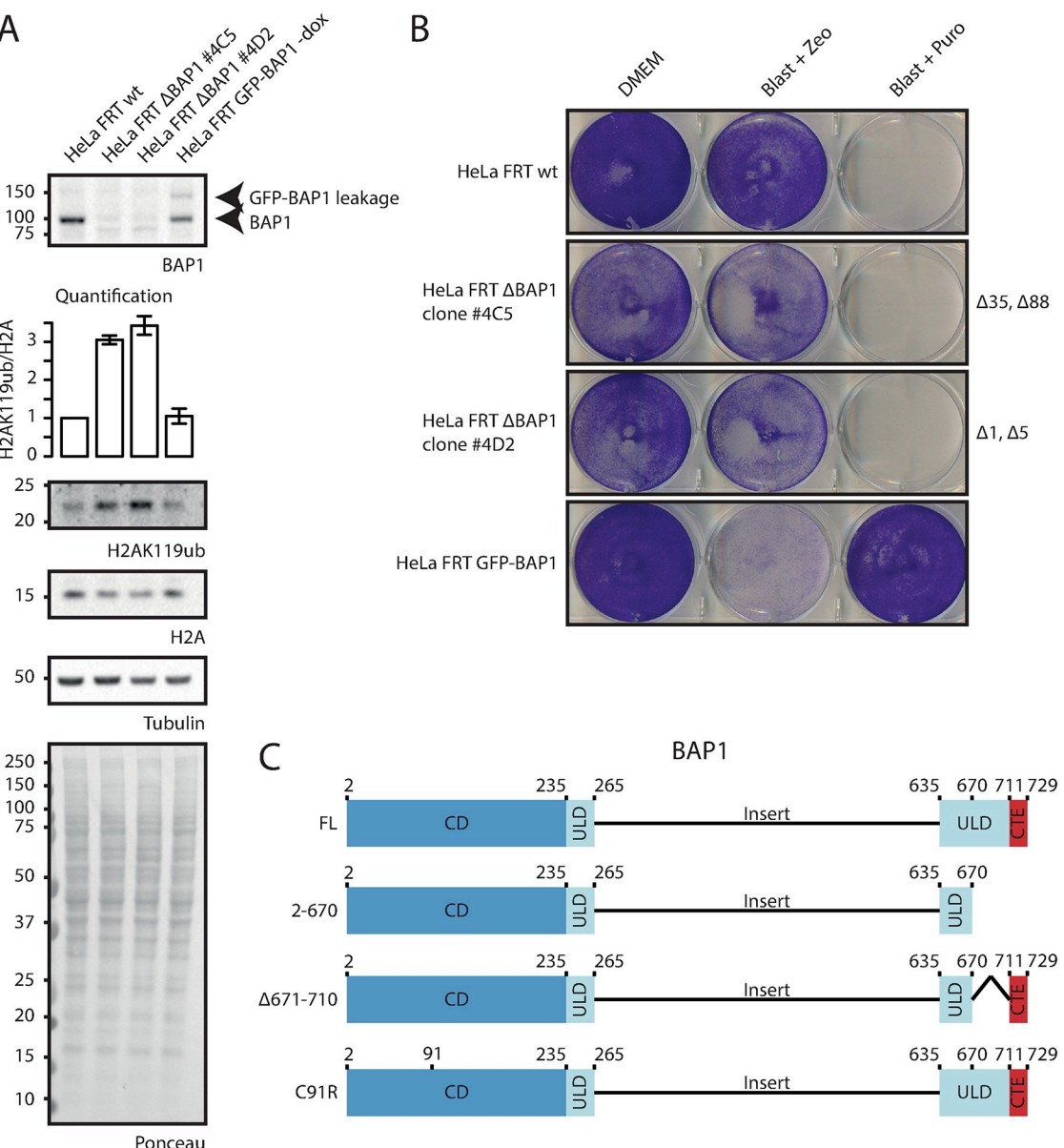

**Fig 2. HeLa FRT ΔBAP1 generation using CRISPR knockout.** (A). Immunoblot analysis and quantification of relative H2AK119ub1 levels of HeLa FRT wild-type cells, two HeLa FRT BAP1 CRISPR clones (HeLa FRT ΔBAP1 #4C5 and #4D2) and HeLa FRT GFP-BAP1 –dox cells. Equal amount of cell lysate was analyzed using the listed antibodies to validate the effect of BAP1 loss (B). Selection marker sensitivity test for the cell lines tested in (A). Equal amounts of cells were seeded in 6-well plates and grown in the listed growth conditions. Cells were fixed and stained using crystal violet staining. (C). BAP1 truncation mutants introduced in the HeLa FRT ΔBAP1 cells.

suppresses IP3 receptors at resting state [39]. Other interacting proteins include transport proteins (KPNA2, KPNB1 and TNPO1), INO80 subunits RUVBL1 and RUVBL2, ER related protein VCP, E3 ligase HUWE1 and RNA polymerase II subunits POLR2B and POLR2E. Furthermore some ribosome subunits are identified. As control, the identified proteins were checked against the CRAPome database but they did not belong to the group of common contaminants [40].

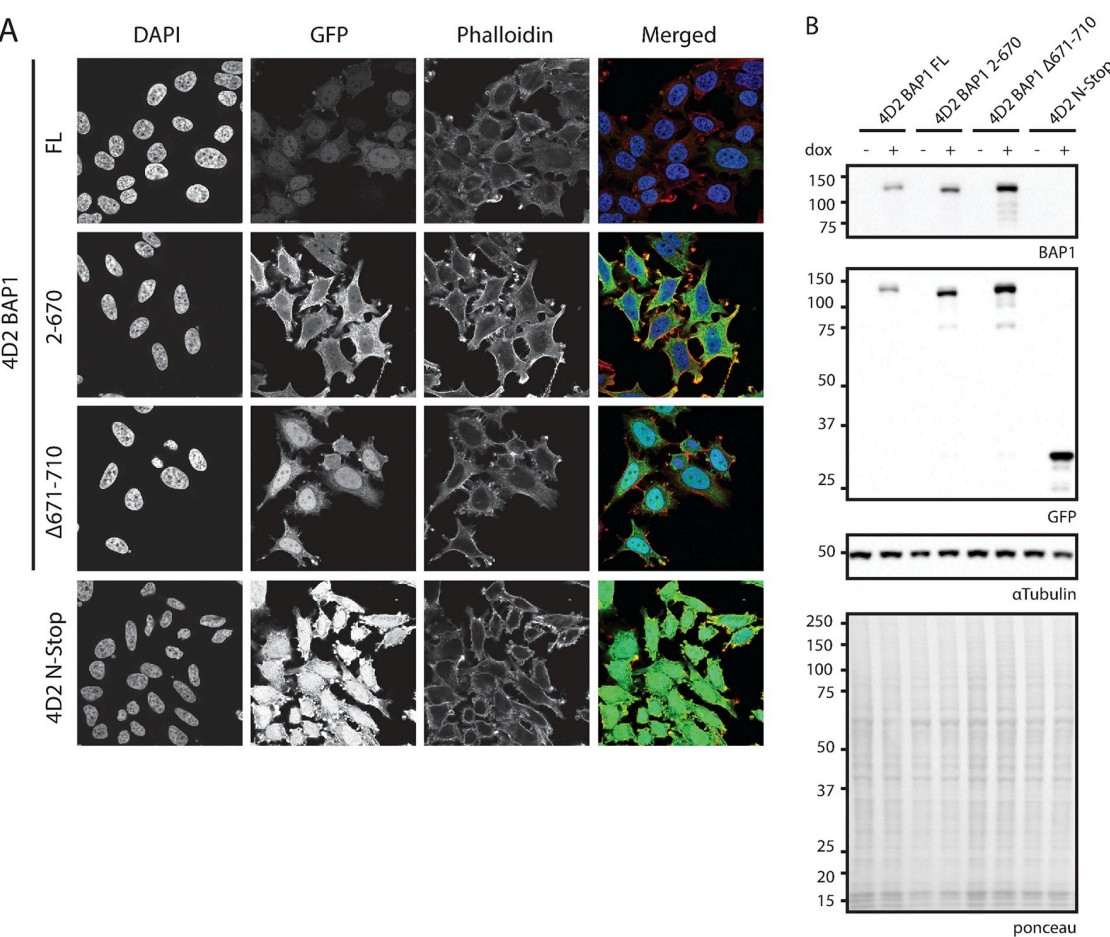

**Fig 3. Localization and expression testing of GFP-BAP1 variants in a ΔBAP1 background.** (A). Confocal microscopy shows difference in localization and expression levels. Listed cell lines were grown on coverslips, stained using DAPI and Phalloidin-633 and mounted on microscopy slides. (B). Immunoblot analysis shows difference in expression levels as observed in (A). Cells were grown in presence of absence of doxycycline and lysates were analyzed using the listed antibodies. N-Stop: HeLa cells expressing only GFP (GFP-TEV-FLAG-3C-Stop linker).

In the cytoplasmic fraction of the 2–670 truncation mutant most of these interactions are lost, indicating that these interactions require the C-terminal region. ASXL proteins that bind the missing ULD domain are absent as expected. Remaining interacting proteins are HCFC1/2, FOXK1/2 and OGT which are known to bind to other regions of BAP1 that are present in this construct [16, 18, 19, 41]. KPNA2, RUVBL1 and CBX3 interactions are still present in the cytoplasm but not in the nuclear fraction as observed before. Interaction with UBE2O is also lost, this corroborates with the UBE2O interaction site that is mapped to the C-terminus of BAP1 and that ubiquitination of BAP1 by UBE2O requires its NLS [26].

The Δ671–710 BAP1 mutant cannot interact with ASXL but has the C-terminal extension including the NLS. Because the C-terminal extension is present, interaction with UBE2O is also restored in the cytoplasmic fraction. Additionally, RBBP7, COPB2, RUVBL1 and CBX3 are again observed. In the nuclear fraction only CBX3, but not UBE2O is observed. Overall the AP-MS experiments identify novel interacting proteins that are found interacting only with full-length BAP1 but not the truncation mutants.

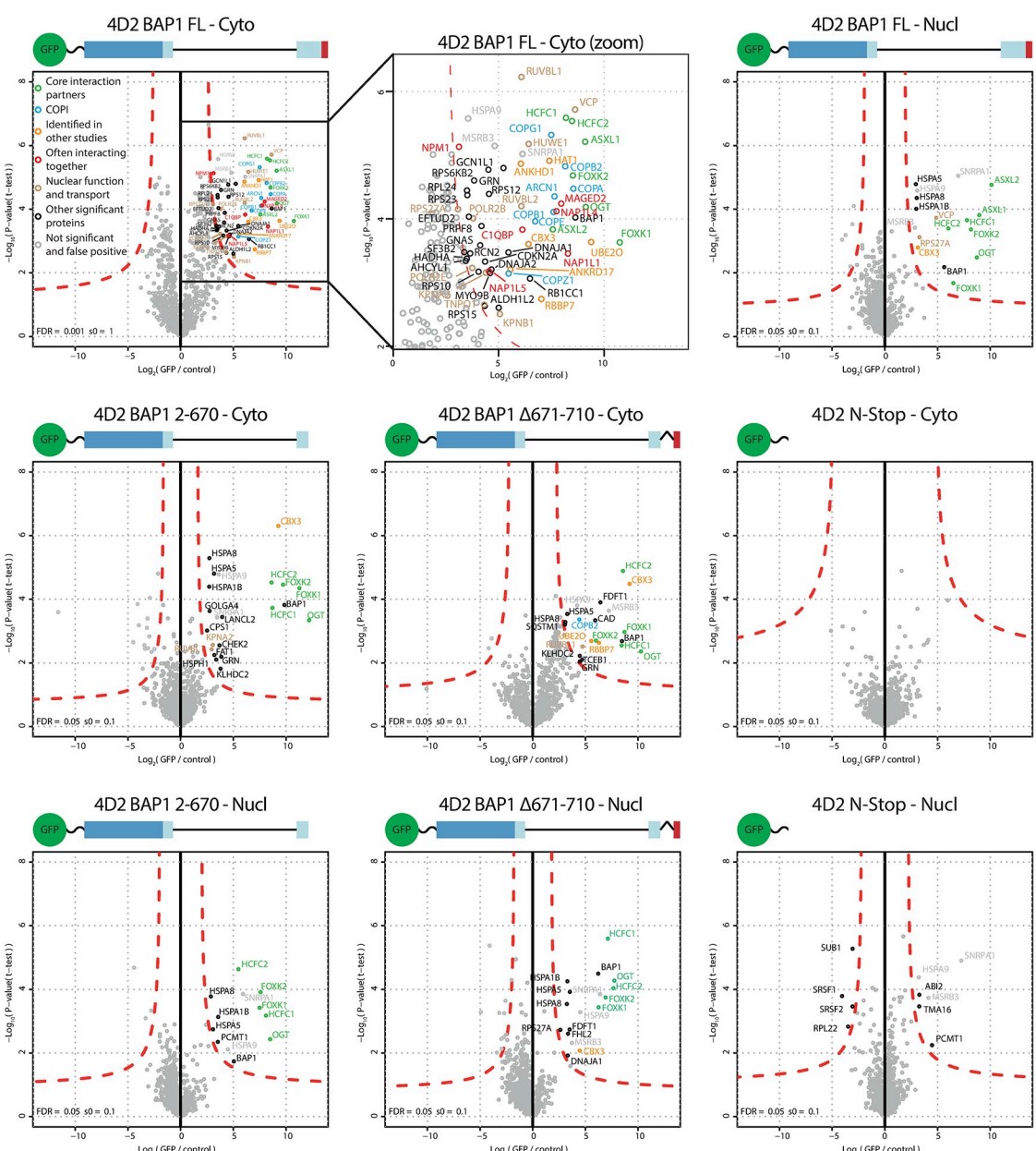

**Fig 4. AP-MS analysis of cytoplasmic and nuclear extracts generated from BAP1 variants in a ΔBAP1 background.** Cytoplasmic and nuclear extracts were generated for the listed cell lines and GFP or mock immunoprecipitation was analyzed using mass spectrometry. Images depict the construct used for immunoprecipitation. Green sphere and wavy line depict the GFP and linker (not to scale to BAP1). N-Stop: HeLa cells expressing only GFP (GFP-TEV-FLAG-3C-Stop linker). Interacting proteins are color coded according to following classification. Green: Core interacting proteins. Blue: COPI subunits. Yellow: BAP1 interacting proteins identified in previous published mass spectrometry experiments. Red: proteins that are often interacting together according to the BioGRID database. Brown: protein with a defined nuclear function. Black: all other interacting proteins. Grey: non-significant and false positive.

## Validation of the BAP1 COPI interaction

The presence of all seven subunits of the heptameric COPI complex in the cytoplasmic interactors of full length BAP1 interested us to validate this interaction. In order to confirm the

interaction between BAP1 and the heptameric COPI complex we performed AP-immunoblot experiments (Fig 5). Purification of BAP1 full length construct via GFP confirms coIP of endogenous COPA on immunoblotting (Fig 5B).

Interestingly, COPA is present as a double band on blot. Presence of DTT in the sample causes the top band to be lost and forces a shift to the lower band, indicating a certain protein mass is lost. This effect is independent of the activity of BAP1, as the catalytic C91R mutant shows the same effect (Fig 5C). Repetition of this experiment in presence of the cysteine DUB inhibitor chloroacetamide suggests that this difference could not be the result of the loss of a ubiquitin moiety via cysteine based DUBs. Overall, the loss of mass of COPA on blot could not be tied to the loss of ubiquitin and additional experiments are required to explain this observation.

To exclude that the BAP1 interaction with the COPI complex is an artifact of overexpression of the exogenous BAP1 construct, we immunoprecipitated endogenous BAP1 using different antibodies against BAP1 in cell lysate from HeLa FRT parental cells (Fig 5D). We noted that the Santa Cruz C-4 antibody has a low immunoprecipitation efficiency. Other monoclonal and polyclonal antibodies raised in either mouse or rabbit have a 13.3–14.3 fold higher efficiency for BAP1 IP. In all BAP1 IP samples, but not the negative control normal IgG samples endogenous COPA coimmunoprecipitates with endogenous BAP1.

We note that only a small fraction of total COPA binds to BAP1 as input levels are much higher than COPA levels in the coIP, correlating with the stoichiometry observed in the AP-MS experiment (Fig 5A and S2 File). AP-immunoblotting of GFP-COPE shows interaction of COPA as expected, however BAP1 cannot be observed which is probably due to the low levels of BAP1 that bind to the COPI complex (Fig 5E). These data confirm the interaction between BAP1 and COPA on endogenous protein and indicate that this interaction is not due to overexpression artefacts.

## The C-terminal KxKxx motif in BAP1 does not mediate COPI binding

COPI complexes have been shown to interact with cargo via distinct C-terminal motifs on the cargo. Canonical COPI binding motifs consists of KKxx and KxKxx and the interaction is mediated by the two charged lysines that bind to the WD-40 repeat of α-COP and β'-COP respectively [30]. Since the C-terminal residues of BAP1 conforms to this motif (KAKRQ) we wanted to test if this tail mediates COPI binding. We mutated the region to SASRQ and introduced it into the 4D2 BAP1 deficient clone. AP-immunoblotting shows no difference in COPA coIP between the BAP1 wt and SASRQ mutants (Fig 5F), indicating that mutation of these two lysines is not enough to disrupt the COPI interaction and suggests that the BAP1 interaction may be mediated via an alternate binding site.

## Discussion

In this manuscript we identify novel cytoplasmic interactors for the tumor suppressor BAP1. This was enabled by removing endogenous BAP1, so that immunoprecipitation of ectopically expressed BAP1 pulled out the entire pool of BAP1 interactors, resulting in noise reduction and better identification of interacting proteins.

The interactome of a protein of interest is defined by the function it has on its interactors. The protein of interest can be activated, or it may be targeted to a location or organelle where its enzymatic activity is needed and it can interact with enzymatic substrates. When and to what extent a certain interactor is binding to the protein of interest is therefore context dependent. It is clear that for BAP1 all these factors apply. ASXL1/2 is required for BAP1 activation [1, 2] and FOXK1/2 and HCFC1 form a ternary complex on target genes to regulate gene

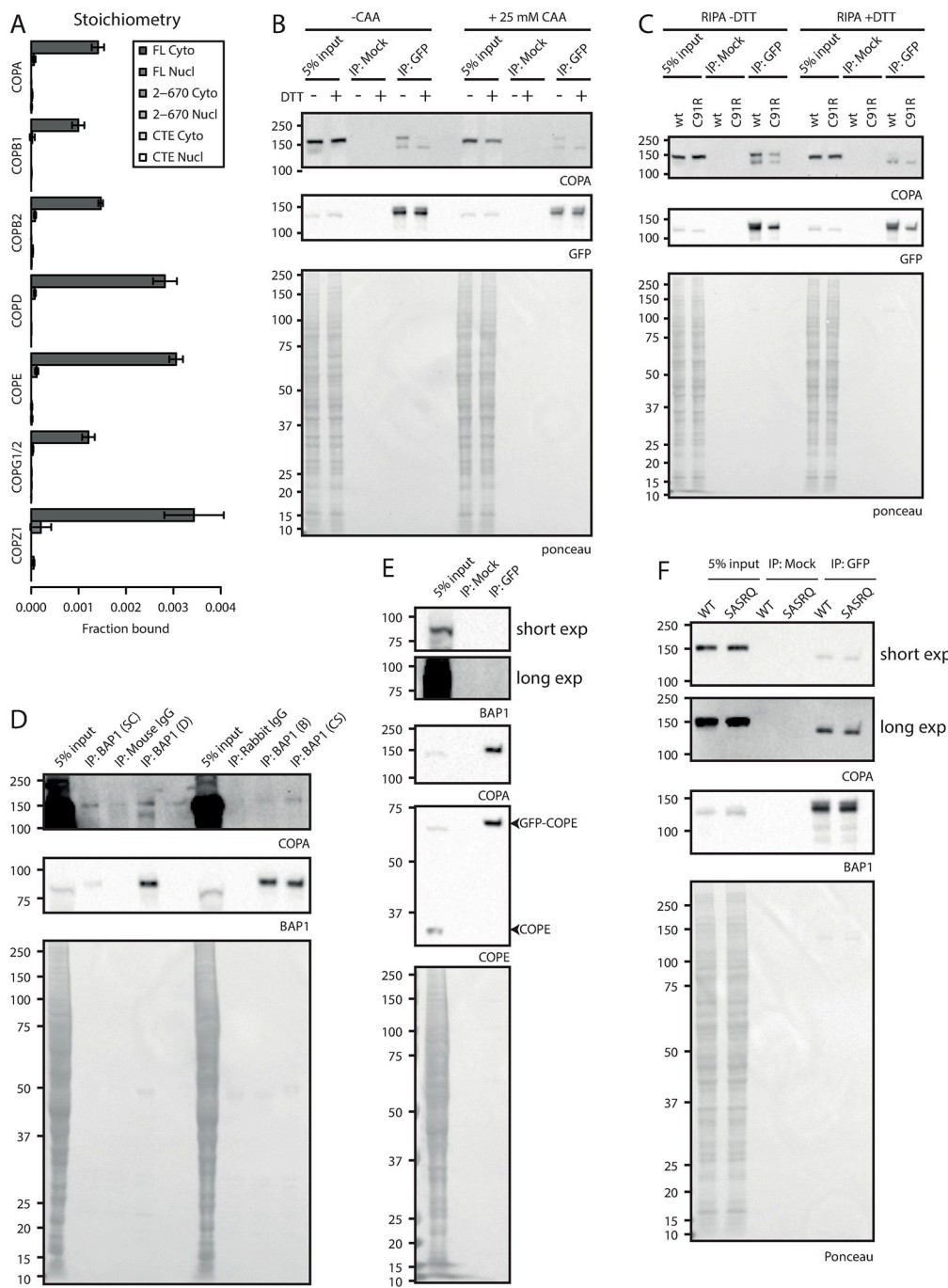

**Fig 5. Characterization and validation of the BAP1 –COPI interaction.** (A). Relative binding of COPI complex subunits: stoichiometry based on label-free quantification of experiment in Fig 4. (B and C). Immunoblot analysis of GFP-BAP1 IP experiments. HeLa cells expressing GFP-BAP1 were grown and lysates were supplemented with CAA or DTT as indicated and used for mock or GFP immunoprecipitation. Blots were analyzed using listed antibodies. (D). BAP1 endogenous IPs using different BAP1 antibodies. HeLa FRT wt cell lysate was used for endogenous immunoprecipitation and blots were analyzed using listed antibodies. (E). Immunoblot analysis of GFP-COPE IP. Cell lysate of GFP-COPE expressing cells was used in GFP immunoprecipitation. Blots were analyzed using listed antibodies. (F). BAP1 C-terminal tail mutational analysis. Cell lysates of GFP-BAP1 wt or SASRQ c-terminally mutated BAP1 was used for mock or GFP immunoprecipitation. Blots were analyzed using listed antibodies.

transcription [41]. BAP1 removes H2AK119ub by its enzymatic activity [1] and therefore needs to be transported to the nucleus to be able to function there, which is likely mediated via transport proteins. Loss of the tumor suppressor BAP1 is linked to development of a distinct set of cancers, making BAP1 an interesting target to study. Since the discovery of its deubiquitination activity towards its main target H2AK119ub [1], logically, many of the studies performed on BAP1 have been targeted towards its nuclear function, while only some have investigated BAP1 in context of its cytoplasmic function. A major fraction of BAP1 protein can be found in the nucleus (Fig 3A), however a significant fraction clearly resides in the cytoplasm of the cell. While relocalization of BAP1 from the nucleus to the cytoplasm leads to sequestering of BAP1 from its nuclear targets [26], its cytoplasmic presence is more than just a physical separation from the nucleus and it has become clear that BAP1 has additional biological functions in the cytoplasm as well [25]. Here we separate the cytoplasmic and nuclear fractions of BAP1 and see distinctive differences in protein binding owing to its location and presence or absence of truncations in the bait protein.

The stoichiometry of BAP1 and ASXL1 molecules in the PR-DUB complex consists of a dimer of BAP1 molecules together with one molecule of ASXL1. When tagging BAP1 for AP-MS experiments, the tagged protein can either form a dimer with another tagged BAP1, or an endogenous BAP1 protein. Additionally, endogenous BAP1 dimers form that cannot be retrieved during affinity purification that also bind to interacting proteins and further diminish the interactome yield of the tagged BAP1. When doing AP-MS experiments, the high-abundant protein interactions like protein subunits of a complex can be easily identified. However when an interaction is low abundant or transient, like enzyme-substrate interactions, a high enough signal needs to be present to be detected. AP-MS on GFP-BAP1 in presence of endogenous protein clearly shows high-abundant interacting proteins like the well-known ASXL, HCFC and FOXK proteins amongst others (Fig 1A and 1B), while some proteins reside at the border of significance. Removal of endogenous BAP1 (Fig 2) allows for recovery of all BAP1 within the cellular extract and boosts identification of the interacting proteins (Fig 4).

BAP1 is a common essential gene [42] that upon removal negatively affects cell fitness. This cell fitness effect is seen in many different cell lines like HCT116 and RPE1, but not present in HeLa cells [43, 44]. Apparently the cell is distinctly rewired, possibly within the PRC2 –PRC1 axis, to allow for BAP1 deletion to be tolerated and thus functional data needs to be examined carefully. This creates the opportunity to use HeLa to study BAP1 in terms of its protein interactions.

The separation of cytoplasm and nucleus allows for identification of protein-protein interaction within those compartments. Proteins that have biological functions in the cytoplasm like the COPI complex are expected to be present in the cytoplasmic fraction (Figs 1A and 4), while proteins that act in the nucleus like MBD6 are expected in the nuclear fraction (Fig 1B). HAT1 and COPI interaction with BAP1 is observed in the cytoplasm and not in the nuclear fractions (Figs 1A and 1B and 4). The interaction with HAT1 in the cytoplasm is interesting because like BAP1 its main function is regulating histone modifications, which raises questions on additional HAT1 functions or substrates.

On the other hand, the COPI interaction corroborates with the function of COPI to form vesicles that are involved in the Golgi to ER transport and intra-Golgi transport. Although the absolute and relative amounts of COPI that interacts with BAP1 in the AP-MS is low (Fig 4 and S2 File), the interaction can still be biologically relevant and is validated with endogenous protein as seen in the immunoblot experiments (Fig 5). Low abundance of interaction can be observed if interaction is weak or transient and such behavior could occur if COPI is a target of BAP1.

COPI cargo binding is facilitated through the C-terminal KKxx or KxKxx sequence that is recognized by the WD-40 motif of α-COP and β'-COP respectively [30]. Mutational analysis of these sequences has shown that mutation of these lysines to serines completely abolishes this interaction [31]. It was intriguing that BAP1 has such a sequence in its C-terminus, but our mutational analysis suggests that the interaction between BAP1 and the COPI complex is not mediated through the C-terminal KxKxx sequence in BAP1. What is interesting is that the interaction with COPB2 is regained if the C-terminal extension of BAP1 is placed back on the construct that is deficient in ASXL binding (Fig 4, Δ671–710 –Cyto). If the COPI complex is an enzymatic substrate of BAP1 then this interaction may be facilitated via a different binding site, possibly involving the interaction with an ubiquitin moiety. Additional experiments will need to be performed to elucidate the binding mechanism.

Over time, novel insights have shown that BAP1 has more functions than just in gene regulation via the deubiquitination of H2AK119ub. Our data have uncovered differential BAP1 partner binding dependent on its physical location outside the nucleus, raising new questions and opportunities with regard to future research on the function of BAP1 and its binding partners.

## Supporting information

**S1 Fig. GFP-BAP1 expression and localization.** (A). Immunoblot analysis of GFP-BAP1 in HeLa FRT cells upon dox induction. Lysates of FRT parental or GFP-BAP1 expressing cells was analyzed on blot using listed antibodies. (B). Confocal microscopy for cell lines used in (A) shows GFP-BAP1 expression and localization in cells. Cells were grown on coverslips and stained using DAPI and Phalloidin-633 and mounted on microscopy slides.
(PDF)

**S2 Fig. AP-MS workflow for GFP-tagged proteins.**
(PDF)

**S3 Fig. Quality control of AP-MS experiment belonging to Fig 1A and 1B.** (A). Histograms of individual mass spectrometry samples. (B). Correlation plots of samples analyzed in (A). Correlation coefficients between log2(LFQ) values of all individual samples within cell lines are depicted as a number (lower triangle) or visually as colored circle (upper triangle).
(PDF)

**S4 Fig. CRISPR clone genotype analysis.** (A and C). TIDE analysis for CRISPR clones 4C5 and 4D2 respectively. (A). The TIDE algorithm was unable to identify the deletions in the 4C5 clone within its search window of -50 to +50 bp. (C). Clone 4D2 was found to contain a 1 and 5bp deletion. (B and D) CRISP-ID analysis for CRISPR clones 4C5 and 4D2 respectively. Top sequence is the used reference sequence. Lower 2 sequences contain the deconvoluted CRISPR clone sequencing results. Colors represent full sequence alignment between reference sequence and deconvoluted sequences. (B). CRISP-ID shows clone 4C5 to contain a 35 and 88bp deletion. (D). Clone 4D2 contains a 1 and 5bp deletion as observed in (C).
(PDF)

**S5 Fig. Cytoplasmic and nuclear separation of BAP1 truncation mutant extracts belonging to Fig 4.** Equal amount of extracts used for mass spectrometry analysis in Fig 4 were blotted and analyzed using the listed antibodies. Tubulin is used as a cytoplasmic protein marker and Abraxas is used as a nuclear protein marker.
(PDF)

**S6 Fig. Quality control of AP-MS experiment belonging to Fig 4.** (A). Histograms of individual mass spectrometry samples. (B).–Correlation plots of samples analyzed in (A).

Correlation coefficients between log2(LFQ) values of all individual samples within cell lines are depicted as a number (lower triangles) or visually as colored circle (upper triangles). (PDF)

**S1 File. Supplementary results.**
(DOCX)

**S2 File. Stoichiometry of interacting proteins identified in Fig 4.**
(XLSX)

**S1 Raw images.**
(PDF)

## Acknowledgments

The authors would like to thank Patrick H.N. Celie for assistance in obtaining purified GFP, Liesbeth Hoekman for running MS and Yvette Stijf-Bultsma for assistance with cellular work.

## Author Contributions

**Conceptualization:** Roy Baas, Titia K. Sixma.

**Data curation:** Roy Baas.

**Formal analysis:** Roy Baas.

**Funding acquisition:** Haico van Attikum, Titia K. Sixma.

**Investigation:** Roy Baas, Fenna J. van der Wal, Onno B. Bleijerveld.

**Supervision:** Haico van Attikum, Titia K. Sixma.

**Validation:** Roy Baas, Fenna J. van der Wal.

**Writing – original draft:** Roy Baas.

**Writing – review & editing:** Fenna J. van der Wal, Haico van Attikum, Titia K. Sixma.

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
