## [Decision Letter · Decision Letter 0]

30 Jun 2021

PONE-D-21-14759

Proteomic analysis identifies novel binding partners of BAP1

PLOS ONE

Dear Dr. Sixma,

Thank you for submitting your manuscript to PLOS ONE. After careful consideration, we feel that it has merit but does not fully meet PLOS ONE’s publication criteria as it currently stands. Therefore, we invite you to submit a revised version of the manuscript that addresses the points raised during the review process.

Please note the major comments 2 and 3 from reviewer 1, as well as the comment from reviewer 2. 

We look forward to receiving your revised manuscript.

Kind regards,

Paul J. Galardy, MD

Academic Editor

PLOS ONE

2. In your Methods section, please provide additional details regarding the cell lines used in your study and ensure you have described the source. For more information regarding PLOS' policy on materials sharing and reporting, see https://journals.plos.org/plosone/s/materials-and-software-sharing#loc-sharing-materials, and for more information on PLOS ONE's guidelines for research using cell lines, see https://journals.plos.org/plosone/s/submission-guidelines#loc-cell-lines.

“The authors would like to thank Patrick H.N. Celie for assistance in obtaining purified GFP, Liesbeth Hoekman for running MS and Yvette Stijf-Bultsma for assistance with cellular work. This work was funded by NWO-ALW OPEN 2015.091, NWO X-omics Initiative, KWF 2015-8082 and Oncode Institute.”

 “KWF Kankerbestrijding (DCS):Titia K Sixma 8082; TKS & HvA; www.kwf.nl

Oncode Institute:TKS;www.oncode.nl

NWO | Aard- en Levenswetenschappen, Nederlandse Organisatie voor Wetenschappelijk Onderzoek (NWO-ALW): 2015.091; TKS; www.nwo.nl

Nederlandse Organisatie voor Wetenschappelijk Onderzoek (NWO): X-omics; NKI; www.nwo.nl

4. Regarding blot/gel data: PLOS ONE now requires that submissions reporting blots or gels include original, uncropped blot/gel image data as a supplement or in a public repository. This is in addition to complying with our image preparation guidelines described at https://journals.plos.org/plosone/s/figures#loc-blot-and-gel-reporting-requirements. These requirements apply both to the main figures and to cropped blot/gel images included in Supporting Information. If the manuscript is positively reviewed, we will ask the authors to provide any missing raw image data for blot/gel results when they submit their first revision. As part of your review, please ensure that figures reporting blot or gel images comply with the journal’s image preparation guidelines and that the original data are provided following the journal’s request.  If you have any questions or concerns about blot/gel figures or data for this submission, please email us at plosone@plos.org before issuing a decision letter.

Please see the comments of the reviewers for details.

Reviewers' comments:

Reviewer's Responses to Questions

**Comments to the Author**

1. Is the manuscript technically sound, and do the data support the conclusions?

Reviewer #1: Yes

Reviewer #2: Yes

2. Has the statistical analysis been performed appropriately and rigorously? 

Reviewer #1: N/A

Reviewer #2: Yes

3. Have the authors made all data underlying the findings in their manuscript fully available?

Reviewer #1: Yes

Reviewer #2: Yes

4. Is the manuscript presented in an intelligible fashion and written in standard English?

Reviewer #1: Yes

Reviewer #2: Yes

5. Review Comments to the Author

Reviewer #1: In the present work Baas et al. conducted a study to identify BAP1 interacting proteins. The authors expressed a tagged version of BAP1 in a cell line lacking its endogenous form, affinity purified it, then identified the interacting partners by mass spectrometry. These interacting partners are: HAT1 and COPI.

The results are very compelling, the data are well presented, the paper I well written.

Major comments:

1. I have to agree with the other reviewers, that the study mostly focusing on the technicality not on mechanistical details. It is clear, that there is interaction between these partners, but the study lacks any details that would pursue the relevance of these interactions.

2. Some of the details needs to be clarified. For example, it took for a while to find that both GFP and FLAG tags were N’ terminal. I’m not really sure that the FLAG was necessary to use. There was only one blot for it (Supplementary Fig 1.), but it didn’t provide any additional information.

3. We must be absolutely sure that the BAP1-COPI interaction is not an artifact. I think beside the IP for the endogenous BAP1, it would be nice to see a negative control when in an IP the GFP-FLAG fused to a non relevant protein. Especially, since for testing whether the C-terminal motif was necessary for the interaction only anti GFP was used. The possibility of an interaction between GFP-FLAG and COPI needs to be ruled out.

Minor comments

1. I would expect more descriptive figure legends. For example, Figure 1C. Instead of just saying: “Immunoblot analysis of GFP-BAP1 IP in HeLa cells show HAT1 as a BAP1 interacting protein.” You kind of have to also describe what you have done in here. I think this is just the title. For example: X protein was expressed in Y cell line. Cell were lysate prepared as it described there. W amount of protein was loaded, and WB carried out for the antibodies indicated…… This is true for most of the figure legends.

2. On the Supplementary Figure 1A, there is a smaller fragment of the BAP1 fusion protein that is discussed in the text. According to the Materials and methods section, the authors have access to four different kind of BAP1 antibodies. Using these, plus GFP and FLAG, I this case could be nailed down, where the fragment is coming from.

Reviewer #2: In this manuscript, Baas et al have described their findings on novel interactors of the UCH family deubiquitinase, Bap1. Although the connection of the DUB with the polycomb repressor complex (PR-DUB) via binding to AsxL proteins is relatively well established, several biological phenomena, especially differential tissue specific effects of Bap1 mutation, seem to suggest the existence of additional functions and interactors of this DUB. In pursuit of these novel interactors, the authors have employed a strategy toward enrichment of Bap1 interactors in absence of the endogenous protein so as to maximize interactome yield. They have designed a number of Bap1 constructs with variation in the C-terminal segment to capture specific interactors of either the cytoplasmic or nuclear form of the protein. The most striking result is the identification of the cytoplasmic COP1 complex as novel Bap1 interactor, a result that may have some interesting implications. It raises the question as to whether Bap1 can regulate ER-Golgi vesicular traffic. The proteomics experiments are well designed and appear to be well executed and validated. The paper is well written with the results clearly and conservatively interpreted. I would like to recommend its publication as such. In terms of a minor concern, I was hoping to find some discussion regarding the new interacting partners (or a member of the Cop1 complex) being substrate for the enzyme. Is the interaction with the Cop1 complex is to modulate its ubiquitination status?

6. PLOS authors have the option to publish the peer review history of their article (what does this mean?). If published, this will include your full peer review and any attached files.

Reviewer #1: No

Reviewer #2: No

---

## [Author Response · Author response to Decision Letter 0]

5 Aug 2021

Response to Reviewers and Journal requirements

Please find below the explanation of our adjustments in response to reviewers and journal requirements

Academic Editor

PLOS ONE

The entire manuscript was checked against the above linked files to meet the style requirements. Figures were separated into individual figure files with the appropriate naming (Fig1, Fig2 etc.) and file format (.eps). Figures were then uploaded to PACE and converted using the tool to .tif files. Supplementary figures, text and files were separated and appropriately named (S1_Fig.pdf etc.).

2. In your Methods section, please provide additional details regarding the cell lines used in your study and ensure you have described the source. For more information regarding PLOS' policy on materials sharing and reporting, see https://journals.plos.org/plosone/s/materials-and-software-sharing#loc-sharing-materials, and for more information on PLOS ONE's guidelines for research using cell lines, see https://journals.plos.org/plosone/s/submission-guidelines#loc-cell-lines.

All used materials will be shared with the scientific community upon reasonable request, additionally our lab regularly deposits plasmids to Addgene. Furthermore, we have included the origin of the cell lines in the “Cell culture” method section.

“The authors would like to thank Patrick H.N. Celie for assistance in obtaining purified GFP, Liesbeth Hoekman for running MS and Yvette Stijf-Bultsma for assistance with cellular work. This work was funded by NWO-ALW OPEN 2015.091, NWO X-omics Initiative, KWF 2015-8082 and Oncode Institute.”

 “KWF Kankerbestrijding (DCS):Titia K Sixma 8082; TKS & HvA; www.kwf.nl

Oncode Institute:TKS;www.oncode.nl

NWO | Aard- en Levenswetenschappen, Nederlandse Organisatie voor Wetenschappelijk Onderzoek (NWO-ALW): 2015.091; TKS; www.nwo.nl

Nederlandse Organisatie voor Wetenschappelijk Onderzoek (NWO): X-omics; NKI; www.nwo.nl

We removed the funding statement from the Acknowledgement section. The updated funding statement should be:

 “KWF Kankerbestrijding (DCS):Titia K Sixma 2015-8082; TKS & HvA; www.kwf.nl

Oncode Institute; TKS; www.oncode.nl

NWO | Aard- en Levenswetenschappen, Nederlandse Organisatie voor Wetenschappelijk Onderzoek (NWO-ALW OPEN): 2015.091; TKS; www.nwo.nl

Nederlandse Organisatie voor Wetenschappelijk Onderzoek (NWO): X-omics Initiative; NKI; www.nwo.nl

We also included this amended statement in the cover letter.

4. Regarding blot/gel data: PLOS ONE now requires that submissions reporting blots or gels include original, uncropped blot/gel image data as a supplement or in a public repository. This is in addition to complying with our image preparation guidelines described at https://journals.plos.org/plosone/s/figures#loc-blot-and-gel-reporting-requirements. These requirements apply both to the main figures and to cropped blot/gel images included in Supporting Information. If the manuscript is positively reviewed, we will ask the authors to provide any missing raw image data for blot/gel results when they submit their first revision. As part of your review, please ensure that figures reporting blot or gel images comply with the journal’s image preparation guidelines and that the original data are provided following the journal’s request. If you have any questions or concerns about blot/gel figures or data for this submission, please email us at plosone@plos.org before issuing a decision letter.

We have included a file named S1_raw_images.pdf which contains all annotated raw blotting images according to the instructions.

No changes were made in the reference section other than formatting according to the above mentioned style requirements (editor point 1). We did not cite any retracted papers in the manuscript.

Please see the comments of the reviewers for details.

Reviewers' comments:

Reviewer's Responses to Questions

Comments to the Author

1. Is the manuscript technically sound, and do the data support the conclusions?

Reviewer #1: Yes

Reviewer #2: Yes

2. Has the statistical analysis been performed appropriately and rigorously? 

Reviewer #1: N/A

Reviewer #2: Yes

3. Have the authors made all data underlying the findings in their manuscript fully available?

Reviewer #1: Yes

Reviewer #2: Yes

4. Is the manuscript presented in an intelligible fashion and written in standard English?

Reviewer #1: Yes

Reviewer #2: Yes

5. Review Comments to the Author

Reviewer #1: In the present work Baas et al. conducted a study to identify BAP1 interacting proteins. The authors expressed a tagged version of BAP1 in a cell line lacking its endogenous form, affinity purified it, then identified the interacting partners by mass spectrometry. These interacting partners are: HAT1 and COPI.

The results are very compelling, the data are well presented, the paper I well written.

Major comments:

1. I have to agree with the other reviewers, that the study mostly focusing on the technicality not on mechanistical details. It is clear, that there is interaction between these partners, but the study lacks any details that would pursue the relevance of these interactions.

Reviewer 1 agrees with reviewers from the previous round and prefers to see further characterization of the interactions and their functional effects. We would have liked to address this but found for COPI that knockdown of these genes is lethal, whereas on the BAP1 side the interactions are mapped to the functionally critical C-terminus, making these experiments technically extremely challenging. These issues, unfortunately, preclude further validation studies at this point. Further investigation is technically and financially beyond the scope and possibilities of this paper.

2. Some of the details needs to be clarified. For example, it took for a while to find that both GFP and FLAG tags were N’ terminal. I’m not really sure that the FLAG was necessary to use. There was only one blot for it (Supplementary Fig 1.), but it didn’t provide any additional information.

The location of the GFP and FLAG tags is stated in the naming of the used constructs as described in the material and method sections. Additionally, an N-terminal GFP molecule is shown in figure 4 (on top of each graph). To emphasize the N-terminal nature of the linker we included the (N) designation at the beginning of the results section when the construct is introduced. We agree that the FLAG tag is only sparingly used in this study. The FLAG tag inclusion was originally designed for a different study and only used in this context to validate the construct.

3. We must be absolutely sure that the BAP1-COPI interaction is not an artifact. I think beside the IP for the endogenous BAP1, it would be nice to see a negative control when in an IP the GFP-FLAG fused to a non relevant protein. Especially, since for testing whether the C-terminal motif was necessary for the interaction only anti GFP was used. The possibility of an interaction between GFP-FLAG and COPI needs to be ruled out.

We agree that controlling for false positive hits is important in this matter. The linker (GFP-FLAG), consists of a substantial 280 amino acids of which 240 form GFP, and 40 form FLAG and some additional linking amino acids. Affinity purification of this linker alone does not yield any COPI subunits as observed in figure 4 (shown in the middle and bottom right). Additionally, affinity purification on other proteins studied by our lab did not identify any COPI subunits (data not shown). A published example can be found in DOI: https://doi.org/10.1016/j.isci.2021.102435 where this GFP-FLAG construct is used to purify tagged USP7. In Supplemental Table S4 of this study, COPA and COPB2 are detected by the mass spectrometer as non-significant background proteins.

Minor comments

1. I would expect more descriptive figure legends. For example, Figure 1C. Instead of just saying: “Immunoblot analysis of GFP-BAP1 IP in HeLa cells show HAT1 as a BAP1 interacting protein.” You kind of have to also describe what you have done in here. I think this is just the title. For example: X protein was expressed in Y cell line. Cell were lysate prepared as it described there. W amount of protein was loaded, and WB carried out for the antibodies indicated…… This is true for most of the figure legends.

We have added the requested information by the reviewer to the figure legends where applicable, both in the main and in the supplemental figure legends.

2. On the Supplementary Figure 1A, there is a smaller fragment of the BAP1 fusion protein that is discussed in the text. According to the Materials and methods section, the authors have access to four different kind of BAP1 antibodies. Using these, plus GFP and FLAG, I this case could be nailed down, where the fragment is coming from.

Supplemental figure 1 was included to show GFP-fusion protein expression by doxycycline induction in the generated cell line. As indicated by the reviewer in point 2, the BAP1 construct is N-terminally tagged with GFP and FLAG. Therefore, the bands present in 1A GFP and FLAG blots contain the N-terminal part of the protein. The upper band is the full-length protein whereas the lower band represents a construct that is partially degraded from the C-terminus (otherwise the protein wouldn’t have been detected on blot). In the BAP1 blot above additional bands are observed in addition to the two identified bands observed in the GFP and FLAG blots. Since these additional bands are not observed in the GFP and FLAG blots they represent N-terminal degradation products. Upon affinity purification these bands are not bound by the GFP affinity purification beads and do not influence the subsequent mass spectrometry analysis.

Reviewer #2: In this manuscript, Baas et al have described their findings on novel interactors of the UCH family deubiquitinase, Bap1. Although the connection of the DUB with the polycomb repressor complex (PR-DUB) via binding to AsxL proteins is relatively well established, several biological phenomena, especially differential tissue specific effects of Bap1 mutation, seem to suggest the existence of additional functions and interactors of this DUB. In pursuit of these novel interactors, the authors have employed a strategy toward enrichment of Bap1 interactors in absence of the endogenous protein so as to maximize interactome yield. They have designed a number of Bap1 constructs with variation in the C-terminal segment to capture specific interactors of either the cytoplasmic or nuclear form of the protein. The most striking result is the identification of the cytoplasmic COP1 complex as novel Bap1 interactor, a result that may have some interesting implications. It raises the question as to whether Bap1 can regulate ER-Golgi vesicular traffic. The proteomics experiments are well designed and appear to be well executed and validated. The paper is well written with the results clearly and conservatively interpreted. I would like to recommend its publication as such. In terms of a minor concern, I was hoping to find some discussion regarding the new interacting partners (or a member of the Cop1 complex) being substrate for the enzyme. Is the interaction with the Cop1 complex is to modulate its ubiquitination status?

We agree with reviewer 2 that the interaction between BAP1 and the COPI complex may be a substrate interaction as evidenced by the low protein stoichiometry observed in the affinity purification (Fig 5). This is discussed in the discussion (“If the COPI complex is an enzymatic substrate of BAP1 then this interaction may be facilitated via a different binding site, possibly involving the interaction with a ubiquitin moiety.”), but clearly more experiments are needed to elucidate this hypothesis.

6. PLOS authors have the option to publish the peer review history of their article (what does this mean?). If published, this will include your full peer review and any attached files.

Do you want your identity to be public for this peer review? For information about this choice, including consent withdrawal, please see our Privacy Policy.

Reviewer #1: No

Reviewer #2: No

Figures were uploaded and converted to the journal standards using PACE.

---

## [Editor Report · Decision Letter 1]

8 Sep 2021

Proteomic analysis identifies novel binding partners of BAP1

PONE-D-21-14759R1

Dear Dr. Sixma,

We’re pleased to inform you that your manuscript has been judged scientifically suitable for publication and will be formally accepted for publication once it meets all outstanding technical requirements.

Kind regards,

Paul J. Galardy, MD

Academic Editor

PLOS ONE

Additional Editor Comments (optional):

Thank you for addressing the comments of the reviewers. I feel that there are no further items requiring clarification.
---

## [Editor Report · Acceptance letter]

22 Sep 2021

PONE-D-21-14759R1 

Proteomic analysis identifies novel binding partners of BAP1 

Dear Dr. Sixma:

I'm pleased to inform you that your manuscript has been deemed suitable for publication in PLOS ONE. Congratulations! Your manuscript is now with our production department. 

Kind regards, 

on behalf of

Dr. Paul J. Galardy 

Academic Editor

PLOS ONE